# The Influence of Different Levels of Sodium Nitrite on the Safety, Oxidative Stability, and Color of Minced Roasted Beef

**Karolina M. Wójciak, Dariusz M. Stasiak *** 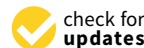 **and Paulina Kęska**

Department of Animal Raw Materials Technology, University of Life Sciences in Lublin, 20-033 Lublin, Poland
* Correspondence: dariusz.stasiak@up.lublin.pl; Tel.: +48-81-462-3343

**Abstract:** This study focuses on collecting actual data on the workable possibility of reducing the technological use of nitrites in beef products according to the present trends in nutrition, especially in terms of European Union (EU) food law. Measurements of safety by technological (pH value, water activity, N-nitrosamine), microbiological, oxidative stability (thiobarbituric acid reactive substances, oxidation-reduction potential), and color parameter (CIE L*a*b*, total heme pigment and heme iron) methods were taken after production and storage. The roasted beef with a reduced inclusion level of sodium nitrite (75 mg/kg and below) was more vulnerable to lipid oxidation. The quantities of primary lipid oxidation products were related to the sodium nitrite inclusion level (50–150 mg/kg). *Clostridium* spp., *Staphylococcus aureus,* and *Listeria monocytogenes* were not detected in any of the samples tested during all the experiments. The total count of *Enterobacteriaceae* increased with the decrease in sodium nitrite content, from log 2.75 cfu/g at the highest to log 6.03 cfu/g at the smallest addition of nitrite. The obtained results revealed that the addition of 100 mg/kg of sodium nitrite would be adequate for minced roasted beef, without significant unexpected effects on color, oxidative stability, and microbiological safety compared with the control (150 mg/kg).

**Keywords:** sodium nitrite; meat products safety; lipid oxidation; pathogen bacteria; N-nitrosamines

## 1. Introduction

Meat products are a unique food form for many reasons. They form the basis of the diet of many people around the world, although there are others who prefer meatless diets for different reasons. Europeans consume a lot of meat. The annual consumption is estimated to be between 60 and 135 kg of meat and meat products (including 30–60 kg of pork) per capita. This is mainly due to the fact that meat is a rich source of protein. Meat products are a rich source of exogenous amino acids (phenylalanine, lysine, leucine, isoleucine, methionine, threonine, tryptophan, and valine) and relatively exogenous precursors (arginine and histidine) for the synthesis of nitrogen compounds of physiological importance, (for example, serotonin, nicotinic acid, carnitine, thyroxine, creatine, heme, and glutathione). They are the basic source of B vitamins (B1, B12, and B2) and vitamins A, E, C, and $B_3$. Meat products are believed to be an essential source of many micro and macro elements (e.g., iron, zinc). However, Püssa [1] showed some toxic substances that unfortunately may be present in meat and meat products, including arsenic, cadmium, lead, polychlorinated biphenyls (PCBs), dioxins, aflatoxins, ochratoxins, ptaquiloside, phytanic acid, dichlorodiphenyltrichloroethane (DDT), leukotoxins, polycyclic aromatic hydrocarbons (PAHs), biogenic amines, botulinum toxin, bisphenol A, phthalates, and nitrites. These toxic substances are from various sources. Nitrites can react with hemoglobin to form methemoglobin, which lacks oxygen transport ability, and produce *N*-nitrosamines by reaction with secondary amines which are capable of mutagenic and carcinogenic actions [1].

Because of the potential risk of formation of carcinogenic substances in meat products, the reduction or total elimination of the addition of sodium/potassium nitrite from food products has been one of the main research areas in the field of meat science worldwide. Today, nitrite is used to meet consumer requirements with respect to product safety (protection against multiplication of *Clostridium botulinum*) and sensory characteristics connected with cured meats. Correspondingly, the process of meat curing has been traditionally coupled with processed meats in order to reach proper color, texture, flavor, safety, and shelf-life characteristics that make the products diverse. Current European Union (EU) regulations concerning the use of nitrite and nitrate differ with respect to both the method of curing used and the product that is cured. According to Directive 2006/52/EC of the European Parliament [2] and of the Council and Commission Regulation (EU) No 1129/2011 [3], all in all, 150 mg nitrite per kg is permitted to be inserted into all meat products plus 150 mg nitrate per kg for non-cooked meat products. Finally, maximum levels of 300 mg nitrite and nitrate per kg can be used for ripening ham production. In cooked meat products, no more than 150 mg nitrite per kg is allowed. However, the maximum allowed nitrite level is 100 mg/kg in the sterilized meat products. When curing meat, nitric oxide is formed from sodium nitrite added to meat. The minimum addition of nitrite for obtaining a visible cured meat color is determined experimentally as approximately 25 mg/kg [4]. According to Sindelar and Milkowski [5], the limit of sodium nitrite in order to obtain the required cured meat flavor and oxidative stability is above 50 mg/kg. The influence of nitrite on the flavor of meats is well described, but the chemical movements are still unclear. Villaverde et al. [6] and Berardo et al. [7] presented the complicated but unclear consequences of curing on meat lipid and protein transformation. In addition, the nitrite promotes the formation of Strecker aldehydes, thus forming the overall sensory quality of cured meats. Therefore, the reduction of nitrite level in meat products should be carefully studied.

Nitrite inhibits the growth of pathogenic bacteria such as *Salmonella enterica serovar Typhimurium*, *Listeria* spp., and *Clostridium botulinum* [8], which under anaerobic condition produces the most lethal neurotoxin. Nitrite in meat (50–150 mg/kg) can increase the pace of the development of various aerobic and anaerobic microorganisms. In January 2016, the Food Chain Evaluation Consortium discussed the requests of European Commission and inferred in the report that a fair inclusion level of 100 ppm of added nitrite would be adequate for the majority of products, without having a significant effect on color, flavor, and microbiological safety [9].

Nitrite is a multifunctional food additive in meat processing. On the one hand, the result of a previous study showed that nitrite use should be limited due to its potential negative influence on human health. On the other hand, some studies revealed the beneficial effect of nitrite on human health [10]. Considering the varied expectations of both food producers and consumers, it is important to pursue the impact of different amounts of sodium nitrite on the safety, oxidative stability, and the color of minced roasted beef to reduce or eliminate toxic substance from meat products. Therefore, the objective of the research was to collect physicochemical and microbiological data on the real demand of nitrites. These data will be useful for achieving advances in meat technology, especially on the ability to limit the use of nitrites according to EU health policy.

## 2. Materials and Methods

### 2.1. Preparation and Storage of Product

For the experiment, the meat (beef and pork back fat) was obtained from a local abattoir. The weight of the young bull (*Limousine*) carcass was approximately 450 kg. The beef and pork back fat (48 h postmortem) without quality defects (dark, firm, and dry—DFD, incorrect sensory quality) was transferred in a cold box to the laboratory, sliced with a knife, and then divided into four equal portions (85% of beef and 15% pork back fat). Different inclusion levels of sodium nitrite were added to the portions of meat. The control sample was enriched with 150 mg/kg of sodium nitrite (N_150). To the other three samples, inclusion levels of 100 mg/kg (N_100), 75 mg/kg (N_75), and 50 mg/kg (N_50) were added, respectively. The addition of salt was 2.2% of the meat mass. The dry curing method was

used. Then, each meat portion (beef and fat) was separately minced (hole diameter 5 mm) and mixed (4 min) together using a universal machine type KU2–3EK (MESKO-AGD, Poland). The homogenous meat stuffing samples (1 kg) in aluminum molds were roasted until reaching the temperature of 72 °C in its geometric center using the digital DT-34 thermometer equipped with an ST-01-1120 probe (TERMOPRODUKT, Bielawa, Poland). The roasting was performed in a convectional XF135 steam oven (UNOX, Italy) at 220 °C. Next, the samples were cooled down with fresh air until they reached the temperature of 20–25 °C. The product was then packed in low-density polyethylene (LDPE) bags using VAC-10DT (EDESA, Palmiry, Poland) and stored in a dark and cold (4 °C) chamber for 21 days. The first sample was tested on the production day (day 0), and the other ones were tested on days 7, 14, and 21.

### 2.2. Analysis of Product Safety

### 2.2.1. Water Activity and pH Value Measurement

The water activity ($a_w$) measurement of product was conducted at 20 °C using the LabMaster-aw water activity analyzer (NOVASINA AG, Lachen, Switzerland).

The pH value was measured potentiometrically. Initially, the homogenates were obtained through homogenizing 10 g of product with 50 mL of cold (4 °C) distilled water for 60 s using the Ultra-Turrax T25 Basic (IKA, Staufen, Germany). The corresponding pH readings were recorded at room temperature with a CPC-501 pH-meter (ELMETRON, Zabrze, Poland) equipped with an ERH-111 combined electrode (HYDROMET, Gliwice, Poland).

### 2.2.2. Microbial Analysis

A 20-g sample of product was homogenized with 180 mL of peptone water for 1 min in a Stomacher Lab-Blender 400 (SEWARD MEDICAL, London, UK). Next, serial dilutions of the sample were made. The lactic acid bacteria (LAB) count was determined according to ISO 15214:2002 [11], and by using MRS agar plates (MERCK, Darmstadt, Germany), LAB were proliferated anaerobically at 30 °C for 72 h. The colonies of *Clostridium* sp. were counted on TSC agar (MERCK, Darmstadt, Germany) incubated anaerobically at 37 °C for 20 ± 2 h according to ISO 7937:2005 [12]. *Staphylococcus aureus* colonies were counted using the method described in ISO 6888–2:2001/A1:2004 [13], while the colonies of *Listeria monocytogenes* and *Enterobacteriaceae* were counted using ISO 11290–2:2000 [14] and ISO 21528-2:2005 [15], respectively. The results are expressed as the logarithm of colony forming unit per one gram (log cfu/g) of product.

### 2.2.3. N-Nitrosamine Content

The volatile nitrosamines were extracted using supercritical fluid extraction following the method recommended by the Food Safety and Inspection Service, Procedure No. CLG-NTR3.01. The extract was then allowed to pass through the silica gel trap, and the nitrosamines from sample were retained. A mixture of ethyl ether in dichloromethane was used to elute the retained substances. The resulting elute was concentrated to a small volume and analyzed through gas chromatography with a thermal energy analyzer towards the detection of nitrosamines.

### 2.3. Analysis of Oxidation Stability

### 2.3.1. Measurement of Thiobarbituric Acid Reactive Substances (TBARS) Values

The extent of lipid oxidation in the product was assessed by measuring the amount of TBA-reactive substances at 0, 7, 14, and 21 day of chilling storage, according to the procedure described by Pikul et al. [16]. The values are expressed as mg of malondialdehyde (MDA) per one kilogram of the sample.

### 2.3.2. Measurement of Oxidation–Reduction Potential (ORP)

The ORP of the product was measured in a prepared homogenate. A mixture of 10 g of the product, 30 mL of cold deionized water, and 50 μL of butylated hydroxytoluene (BHT, 7.2% in ethanol) was homogenized for 1 min at room temperature using the Ultra-Turrax T25 Basic. The addition of BHT protected the blended sample against oxidation. The ORP was assessed in mV using a pH meter equipped with redox platinum electrode type ERPt-13 (Hydromet, Gliwice, Poland). The readings were noted 5 min after immersing the electrode in the mixture.

### 2.4. Analysis of Color

### 2.4.1. Total Pigments (OZB) and Heme Iron (Fe) Analyses

The meat pigments in the product were determined using a spectrophotometric method as described by Hornsey [17]. Ferric chloride heme (i.e., hemin) in meat extract was estimated by absorption spectroscopy of 640 nm using a UV-Vis spectrophotometer (Nicolet Evolution 300, Thermo Electron Corp., USA). The heme iron amount was calculated following the procedure depicted by Clark et al. [18].

### 2.4.2. Product Color Evaluation

The spectrophotometer series 8200 (X-RITE Inc., USA) was used to determine the color parameters of the product. The measurement was taken on the viewport set at 13 mm, illuminant D65 (daylight, 6500K), and 10° standard observer. Color parameters were determined using the CIELAB color space [19] which describes color as three numerical values, namely L* for the lightness and a* and b* for the green–red and blue–yellow color components, respectively. The total color variety (ΔE) between pattern and the sample was calculated according to the following equation:

$$\Delta E = \sqrt{\left(L_{pattern} - L_{sample}\right)^2 + \left(a_{pattern} - a_{sample}\right)^2 + \left(b_{pattern} - b_{sample}\right)^2} \tag{1}$$

The apparatus was warmed up and calibrated before tests with the original X-Rite white and black standards. The measurements were conducted at room temperature of six diverse independent areas on freshly cut slices being 10 mm thick. Finally, the total color difference (ΔE) between the pattern (control sample, N_150) and the remaining samples (i.e., N_100, N_75, and N_50) was calculated.

### 2.5. Statistical Procedure

The study was randomized and repeated for three times, and every sample was measured triplicate to obtain a strong final signal-to-noise ratio. The acquired data were processed using analytic software Statistica® (StatSoft Poland). Statistical and multivariate analyses (cluster analysis, CA) were carried out. With the use of the Tukey's range *t*-test, significant differences between mean values were calculated at the $p < 0.05$ level. Furthermore, Pearson's correlation coefficient (r) among the selected predictors was determined. Prior to the chemometric method, the data matrix was auto-scaled. The dataset was treated by Ward's method of linkage with squared Euclidean distance as a measure of similarity for multivariate analysis. A hierarchical method was selected for clustering analysis, which includes measurement of similarity between objects (different variants), and samples with maximal similarities were grouped together graphically on dendrograms.

## 3. Results

### 3.1. Determination of Product Safety

Sodium nitrite inclusion level and storage time had statistically significant ($p < 0.001$) effects on pH and water activity of samples in groups. The application of half and two-thirds of the maximum permitted nitrite inclusion level inclusion level significantly reduced the initial pH values of beef

during the storage period (Table 1). The reduction of sodium nitrite affected the acidity of roasted, minced beef samples (Table 1). A significantly higher acidity was observed in the N_75 and N_50 samples after production and during storage period (6.07–6.21). Samples N_150 and N_100 had significantly lower acidity than the other samples after production and during storage period. The pH value was systematically increased in the N_150 and N_100 samples during storage. For N_75 and N_50 samples, the situation was reversed. During storage, a gradual decrease in the pH value was observed.

**Table 1.** pH value and water activity of roasted beef during storage (n = 9).

| Parameter | Sample | Day 0 | Day 7 | Day 14 | Day 21 |
|---|---|---|---|---|---|
| pH value | N_150 | 6.29 ± 0.01 [bA] | 6.25 ± 0.03 [cA] | 6.23 ± 0.01 [dA] | 6.31 ± 0.01 [aA] |
| | N_100 | 6.27 ± 0.01 [aA] | 6.26 ± 0.01 [bA] | 6.17 ± 0.01 [cB] | 6.28 ± 0.01 [abB] |
| | N_75 | 6.17 ± 0.02 [aC] | 6.16 ± 0.02 [aC] | 6.07 ± 0.01 [cD] | 6.11 ± 0.04 [bD] |
| | N_50 | 6.21 ± 0.01 [aB] | 6.18 ± 0.01 [bB] | 6.08 ± 0.02 [dC] | 6.16 ± 0.01 [cC] |
| Water activity | N_150 | 0.969 ± 0.00 [bA] | 0.97 ± 0.00 [bB] | 0.975 ± 0.00 [aB] | 0.975 ± 0.00 [aB] |
| | N_100 | 0.970 ± 0.00 [bA] | 0.971 ± 0.00 [bAB] | 0.975 ± 0.00 [aB] | 0.975 ± 0.00 [aB] |
| | N_75 | 0.970 ± 0.00 [bA] | 0.970 ± 0.00 [bB] | 0.979 ± 0 [aA] | 0.979 ± 0 [aA] |
| | N_50 | 0.970 ± 0.00 [bA] | 0.972 ± 0.00 [bB] | 0.977 ± 0.00 [aB] | 0.977 ± 0.00 [aB] |

Notes: Results are presented as mean ± SD (standard deviation), n – number of samples. Sample: N_150—control with 150 mg/kg of sodium nitrite; N_100—sample with 100 mg/kg of sodium nitrite; N_75—sample with 75 mg/kg of sodium nitrite; N_50—sample with 50 mg/kg of sodium nitrite. Means with capital letters are significantly different ($p < 0.05$) in the same column. Means with small letters are significantly different ($p < 0.05$) in the same row.

Table 1 presents the water activity measurement data of the roasted beef samples during storage. It was observed that sodium nitrite inclusion level and storage time had a statistically significant ($p < 0.001$) effect on water activity. Considering the inclusion level, a significantly higher water activity was observed in N_75 sample than in other samples on days 14 and 21 of chilling storage. Storage time also affected water activity. A gradual increase was observed in the water activity with significant differences between the tested times. Furthermore, the interaction of inclusion level and storage time was significant ($p < 0.001$) for the water activity parameter. In the present study, the yield of *N*-nitrosamines (*N*-nitrosodibutylamine, *N*-nitrosodiethylamine, *N*-nitrosodimethylamine, *N*-nitrosodipropylamine, *N*-nitrosomorpholine, and *N*-nitrosopiperidine) was below the limits of detection (0.5 μg/kg) after production as well as after 21 days of chilling storage. Microbiological assessment did not show the presence of the pathogenic bacteria *Clostridium* sp., *S. aureus*, and *L. monocytogenes* after production and 21 days of storage. For this reason, they formed unique detecting groups (clusters) as illustrated in the dendrogram (Figure 1).

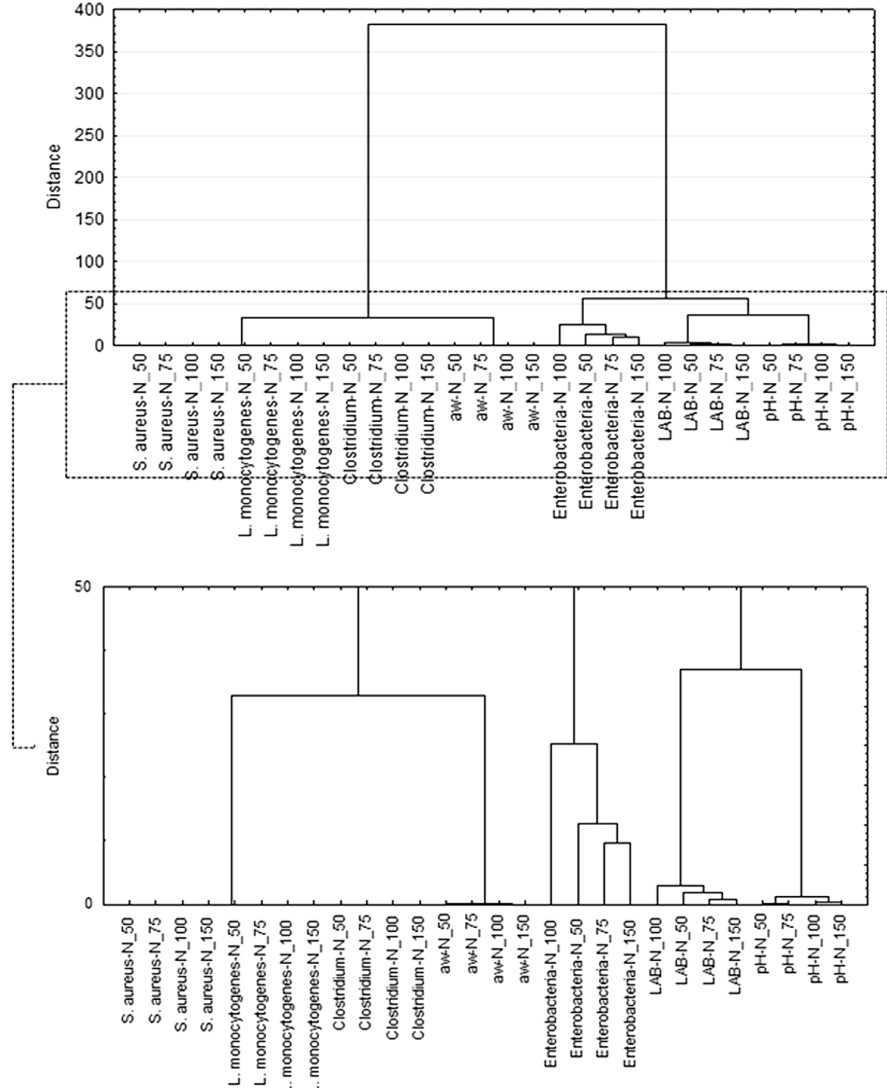

**Figure 1.** Cluster analysis dendrogram of the product safety factors (bacteria: S—*Staphylococcus*, L—*Listeria*, LAB—lactic acid bacteria, pH value, water activity aw; sample: N_150—control with 150 mg/kg of sodium nitrite; N_100—sample with 100 mg/kg of sodium nitrite; N_75—sample with 75 mg/kg of sodium nitrite; N_50—sample with 50 mg/kg of sodium nitrite).

A separate cluster was formed by *Enterobacteriaceae* and LAB, whose presence was additionally strongly correlated with the pH values (they remain within the same sub-cluster) determined in this study. High presence of LAB was observed for all studied samples after roasted beef preparation (log 7.89 cfu/g, log 8.20 cfu/g, log 7.80 cfu/g, and log 8.02 cfu/g for the N_150, N_100, N_75, and N_50 samples, respectively). A slight increase in LAB count was noted for all studied samples after 21 days of storage (log 8.10 cfu/g, log 8.11 cfu/g, log 8.18 cfu/g, and log 8.11 cfu/g for N_150, N_100, N_75, and N_50 sample, respectively). For lower inclusion levels of nitrite samples after production, the growth of *Enterobacteriaceae* was higher than that of the control (N_150). The *Enterobacteriaceae* counts were log 6.35 cfu/g, log 6.46 cfu/g, log 6.55 cfu/g, and log 6.97 cfu/g for N_150, N_100, N_75, and N_50, respectively. After 21 days of chilling storage, the growth of *Enterobacteriaceae* was inhibited in all study samples except the N_50 sample. The *Enterobacteriaceae* count increased in all samples in the following order: N_150 (log 2.75 cfu/g) < N_100 (log 3.86 cfu/g) < N_75 (log 5.14 cfu/g) < N_50 (log 6.03 cfu/g).

### 3.2. Determination of Color Parameters

The red color of cured meat products is one of the important effects of nitrite in meat products. Nitric oxide (NO) is formed when sodium nitrite reacts with the iron present in both myoglobin and metmyoglobin to form pigments and color of cured meat, and the complex formed is called nitrosylmyoglobin. On heating nitrosylmyoglobin, the protein part of heme pigment is denatured, but the red nitrosyl-porphyrin ring system (often called nitrosomyochromogen) still exists even after heating to 120 °C [20].

Only sodium nitrite dose had statistically significant ($p < 0.001$) effect on lightness, yellowness of samples (Table 2) and other sources of variability did not affect the color parameters. Table 2 reports the effect of different inclusion levels of sodium nitrite (from 50 to 150 mg/kg) and storage time on color parameters (L* and b*), total heme pigment (OZB), and heme iron (Fe) content of the roasted beef. The nitrite treatment did not affect ($p > 0.05$) color (a*) parameter of all study samples neither after production nor during the storage period. The a* parameter varied between 14.11 for N_75 and 15.65 for N_150.

**Table 2.** Parameters of color (CIE L*a*b*, total heme pigment OZB, heme iron Fe) of roasted beef during storage ($n = 9$).

| Parameter | Sample | Day 0 | Day 7 | Day 14 | Day 21 |
|---|---|---|---|---|---|
| Lightness | N_150 | 56.22 ± 0.5 [aB] | 56.04 ± 0.61 [aB] | 56.00 ± 1.09 [aA] | 57.71 ± 1.30 [aA] |
| L* | N_100 | 55.61 ± 0.98 [aB] | 55.66 ± 0.51 [aB] | 55.70 ± 0.58 [aA] | 57.16 ± 0.56 [aAB] |
| | N_75 | 56.82 ± 0.72 [aAB] | 57.56 ± 0.71 [aA] | 56.74 ± 1.05 [aA] | 57.78 ± 0.77 [Aa] |
| | N_50 | 55.21 ± 1.12 [aBC] | 55.97 ± 1.22 [aB] | 55.96 ± 1.16 [aA] | 55.76 ± 1.07 [Ba] |
| Redness | N_150 | 15.65 ± 0.71 [aA] | 15.16 ± 0.32 [aA] | 14.61 ± 0.66 [aA] | 14.77 ± 0.69 [aA] |
| a* | N_100 | 14.80 ± 0.42 [aA] | 14.89 ± 0.50 [aA] | 14.61 ± 0.40 [aA] | 14.5 ± 0.49 [aA] |
| | N_75 | 14.66 ± 0.33 [aA] | 14.11 ± 0.18 [aA] | 14.7 ± 0.36 [aA] | 14.86 ± 0.52 [aA] |
| | N_50 | 14.97 ± 0.69 [aA] | 15.04 ± 0.34 [aA] | 15.14 ± 0.44 [aA] | 14.79 ± 0.27 [aA] |
| Yellowness | N_150 | 10.30 ± 0.58 [aA] | 9.69 ± 0.39 [aA] | 9.79 ± 0.38 [aB] | 9.88 ± 0.65 [aA] |
| b* | N_100 | 9.78 ± 0.71 [aA] | 9.30 ± 0.28 [aAB] | 9.38 ± 0.26B [aC] | 9.19 ± 0.37 [aA] |
| | N_75 | 9.85 ± 0.44 [aA] | 9.05 ± 0.21 [aB] | 9.56 ± 0.28 [aB] | 9.83 ± 0.53 [aA] |
| | N_50 | 9.64 ± 0.59 [aA] | 9.62 ± 0.17 [aA] | 10.00 ± 0.29 [aAB] | 9.47 ± 0.30 [aA] |
| OZB | N_150 | 177.91 ± 3.83 [aB] | 176.63 ± 3.08 [abB] | 171.45 ± 7.14 [bB] | 174.59 ± 3.18 [abC] |
| (mg/kg) | N_100 | 183.01 ± 6.13 [bAB] | 187.09 ± 2.63 [abA] | 177.4 ± 3.55B [cbA] | 183.18 ± 3.43 [bB] |
| | N_75 | 174.68 ± 3.18 [aBC] | 175.7 ± 5.06 [aB] | 173.57 ± 3.63 [aB] | 175.61 ± 5.89 [aC] |
| | N_50 | 181.56 ± 8.7 [bB] | 180.2 ± 4.22 [bB] | 181.48 ± 3.36 [bA] | 196.945 ± 5.87 [aA] |
| Fe | N_150 | 15.69 ± 0.34 [aA] | 15.58 ± 0.27 [abB] | 15.12 ± 0.63 [bB] | 15.4 ± 0.28 [abC] |
| (mg/kg) | N_100 | 16.14 ± 0.54 [cA] | 16.5 ± 0.23 [bcA] | 15.64 ± 0.31 [acAB] | 16.15 ± 0.3 [cB] |
| | N_75 | 15.4 ± 0.28 [aAB] | 15.49 ± 0.45 [aA] | 15.31 ± 0.32 [aB] | 15.49 ± 0.52 [aC] |
| | N_50 | 16.01 ± 0.77 [bA] | 15.89 ± 0.37 [bB] | 16 ± 0.56 [bA] | 17.38 ± 0.52 [aA] |

Notes: Results are presented as mean ± SD (standard deviation), n – number of samples. Sample: N_150—control with 150 mg/kg of sodium nitrite; N_100—sample with 100 mg/kg of sodium nitrite; N_75—sample with 75 mg/kg of sodium nitrite; N_50—sample with 50 mg/kg of sodium nitrite. Means with capital letters are significantly different ($p <0.05$) in the same column. Means with small letters are significantly different ($p <0.05$) in the same row.

The different levels of sodium nitrite had significant ($p < 0.05$) effects on L* and b* parameters immediately after 7 and 14 days of storage period. The all sources (i.e. sodium nitrite inclusion level, storage time and interaction) had statistically significant ($p < 0.001$) effects on the total heme pigment and heme iron content. Moreover, as shown in Figure 2, the OZB was at a constant level during 21 days of storage period and formed a separate detecting group.

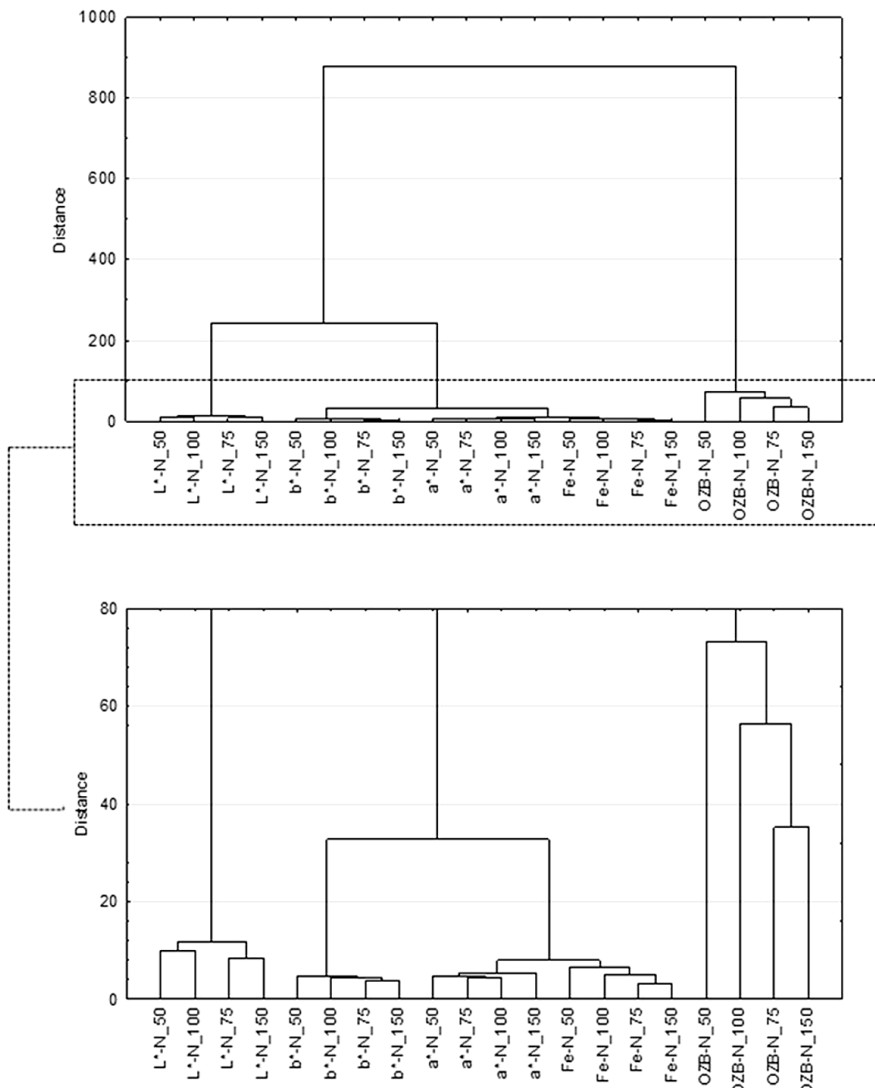

**Figure 2.** Cluster analysis dendrogram of the product color parameters (lightness L*, redness a*, yellowness b*, heme iron Fe, total heme pigment OZB; sample: N_150—control with 150 mg/kg of sodium nitrite; N_100—sample with 100 mg/kg of sodium nitrite; N_75—sample with 75 mg/kg of sodium nitrite; N_50—sample with 50 mg/kg of sodium nitrite).

Except for N_150, the total heme content significantly ($p < 0.001$) increased at the end of storage period. In the literature, we found that a very little amount of nitrite (from 2 to 14 mg/kg) is sufficient to develop reddish-pink color throughout the meat products' shelf-life.

### 3.3. Determination of Oxidative Stability

Another property of nitrite is its ability to retard the oxidation process during storage period and the subsequent warmed-over and rancid flavors developed during thermal processing of meat and meat products. Table 3 shows the effect of different inclusion levels of sodium nitrite (from 50 to 150 mg/kg) and storage time on the oxidative stability of the roasted beef. All sources of variation had a statistically significant ($p < 0.001$) effect on ORP and storage time had no effect on TBARS. A graphical interpretation of the obtained results is shown in the dendrogram (Figure 3).

**Table 3.** Oxidative stability (thiobarbituric acid reactive substances TBARS, oxidation-reduction potential ORP) of roasted beef during storage (*n* = 9).

| Parameter | Sample | Day 0 | Day 7 | Day 14 | Day 21 |
|---|---|---|---|---|---|
| TBARS | N_150 | 0.716 ± 0.05 bB | 0.711 ± 0.03 bD | 0.91 ± 0.13 aB | 0.945 ± 0.07 aB |
| (mg/kg) | N_100 | 0.999 ± 0.09 bB | 0.995 ± 0.04 bC | 0.98 ± 0.05 bAB | 1.158 ± 0.13 aA |
| | N_75 | 0.955 ± 0.06 bB | 1.125 ± 0.07 aB | 1.164 ± 0.21 aAB | 1.12 ± 0.07 aA |
| | N_50 | 2.094 ± 0.71 aA | 1.823 ± 0.05 aA | 1.204 ± 0.34 bA | 1.121 ± 0.04 bA |
| ORP | N_150 | 275.98 ± 6.32 bB | 263.09 ± 10.77 cA | 330.81 ± 1.19 aC | 251.24 ± 1.38 dD |
| (mV) | N_100 | 272.41 ± 6.51 bB | 239.69 ± 5.96 dC | 330.39 ± 3.44 aC | 262.74 ± 5.26 cC |
| | N_75 | 284.7 ± 6.49 BbA | 243.36 ± 4.19 cCB | 342.73 ± 4.06 aA | 284.09 ± 9.02 bB |
| | N_50 | 293.98 ± 18.71 bA | 249.48 ± 3.14 cB | 336.81 ± 4.62 aB | 300.46 ± 10.73 bA |

Notes: Results are presented as mean ± SD ( standard deviation), n – number of samples. Sample: N_150—control with 150 mg/kg of sodium nitrite, N_100—sample with 100 mg/kg of sodium nitrite, N_75—sample with 75 mg/kg of sodium nitrite, N_50—sample with 50 mg/kg of sodium nitrite. Means with capital letters are significantly different (*p* < 0.05) in the same column. Means with small letters are significantly different (*p* < 0.05) in the same row.

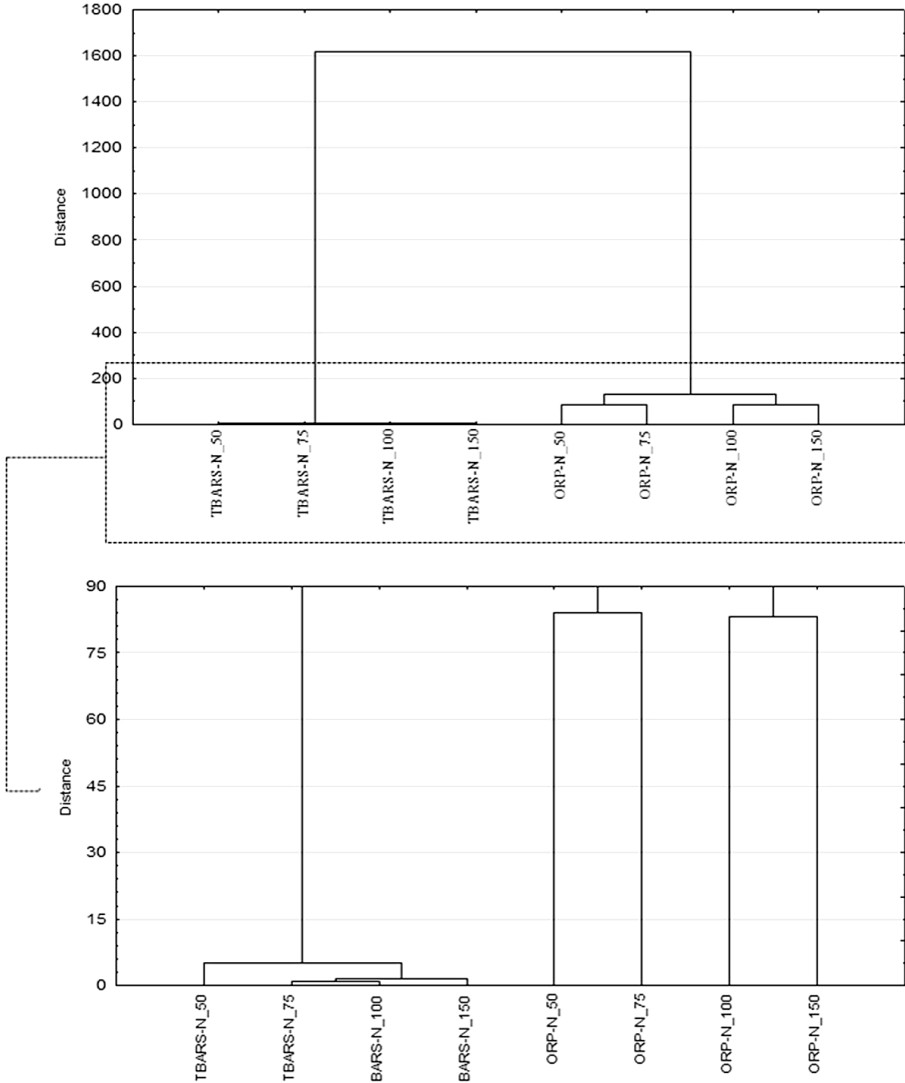

**Figure 3.** Cluster analysis dendrogram of the product oxidation stability (oxidation-reduction potential ORP, thiobarbituric acid reactive substances TBARS; sample: N_150—control with 150 mg/kg of sodium nitrite; N_100—sample with 100 mg/kg of sodium nitrite; N_75—sample with 75 mg/kg of sodium nitrite; N_50—sample with 50 mg/kg of sodium nitrite).

All samples from each variant are grouped, showing no overlap between products with different levels of oxidation stability, and all objects are grouped into two distinct clusters. In the present study, the samples N_75 and N_50 were more susceptible to lipid peroxidation than N_150 and N_100 and had higher ORP. Variants N_150 and N_100 together formed a disjointed sub-cluster within both specified groups of samples. Furthermore, the quantification of secondary products of lipid peroxidation by the TBARS method showed an increase in lipid oxidation products related to sodium nitrite inclusion level in the roasted beef (N_50 > N_75 > N_100 > N_150). The antioxidant effect of nitrite was reported at a level as low as 40 ppm. A significantly higher ORP was observed for samples N_75 and N_50 than for samples (N_150 and N_100) throughout the storage period. Samples treated with higher inclusion level of sodium nitrite (100–150 mg/kg) had decreased potential redox values compared to other samples by ≈18 mV (Table 3). In all study samples, a gradual decrease ($p < 0.05$) was shown at 7 days of storage, followed by a slight increase at 14 and 21 days of storage. Nitrite can form another antioxidant compound, for example, S-nitrosocysteine. In the present study, as pH decreased (Table 1), the ORP increased (Table 3). The highest ORP values (N_75 and N_50) indicated that heme pigments in the ferric state had not stabilized the color of product, which confirmed the results of the ΔE (Figure 4). Therefore, reducing the nitrite inclusion level by half and two-thirds of the maximum permitted inclusion level may cause a significant deterioration of color. Differences between the experimental (N_50, N_75, and N_100) and control trials (N_150) were also observed. Differences in the color between the N_150 and N_100 samples ranged from 1.41 to 1.80 and ranged between one and two units, which according to the International Commission on Illumination (CIE) suggested that they were visible only to the trained eye (differentiates color nuances). For N_75 and N_50, the differences in color compared to the control (N_150) were classified as medium recognizable by a non-specialist in differentiating colors. It was observed that the lower the addition of nitrite, the higher was the difference in the color between the samples and control.

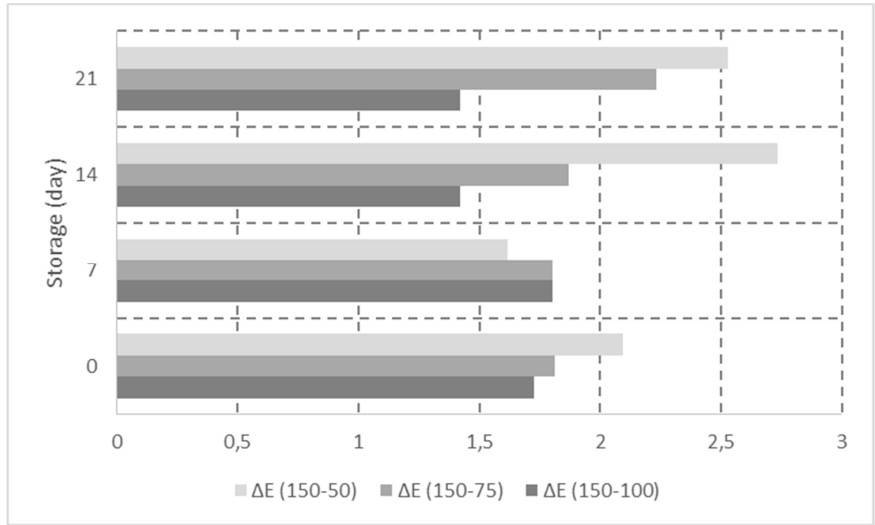

**Figure 4.** Total color difference (ΔE) between roasted beef during storage: (150–50)—N_150 and N_50; (150–75)—N_150 and N_75; (150–100)—N_150 and N-100.

As observed the figure, the dendrogram shows three clusters (Figure 5). The first cluster mainly included N_150 and N_100; the second cluster, N_50; and the third cluster, N_75. The N_150 and N_100 samples were distributed with a low linkage distance value, which could be due to the presence of strong similarities among their properties (color, oxidative stability, and safety). However, the N_75 sample showed the highest linkage distance value, which indicates a more significant difference in relation to other sample properties. As shown in Figure 5, all objects are organized into a dendrogram whose branches are the desired clusters.

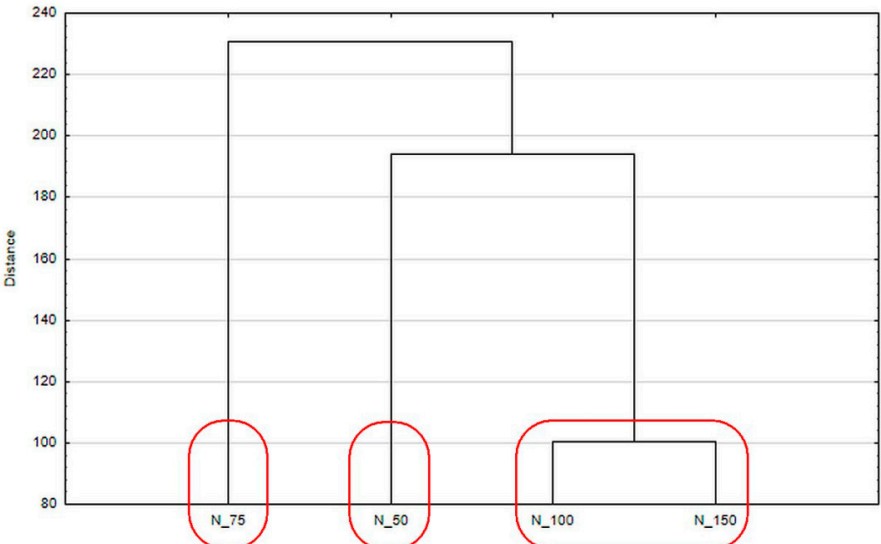

**Figure 5.** Cluster analysis dendrogram of different sodium nitrite addition (Sample: N_150—control with 150 mg/kg of sodium nitrite; N_100—sample with 100 mg/kg of sodium nitrite; N_75—sample with 75 mg/kg of sodium nitrite; N_50—sample with 50 mg/kg of sodium nitrite)

The most spatially remote (least related objects) were presented as clusters represented by variant N_75, and this indicates a more significant difference in relation to the other sample properties. The remaining variants form a separate cluster, in which the most similar results were achieved for variants N_100 and N_150 (forming a common sub-cluster with a low linkage distance value); this finding could be due to the presence of strong similarities among their properties (color, oxidative stability, and safety). In the experiment conducted, a 3D chart was constructed on the basis of the obtained experimental surface. In the models, the *z*-axis represents the OZB (as a color indicator conditioning the choice of the consumer) as a function of TBARS values (*y*-axis) and ORP (*x*-axis) of minced, roasted beef as predictors of oxidation processes (Figure 6A). The relationship between OZB and the ORP value was shown (r = −0.650, *p* > 0.05).

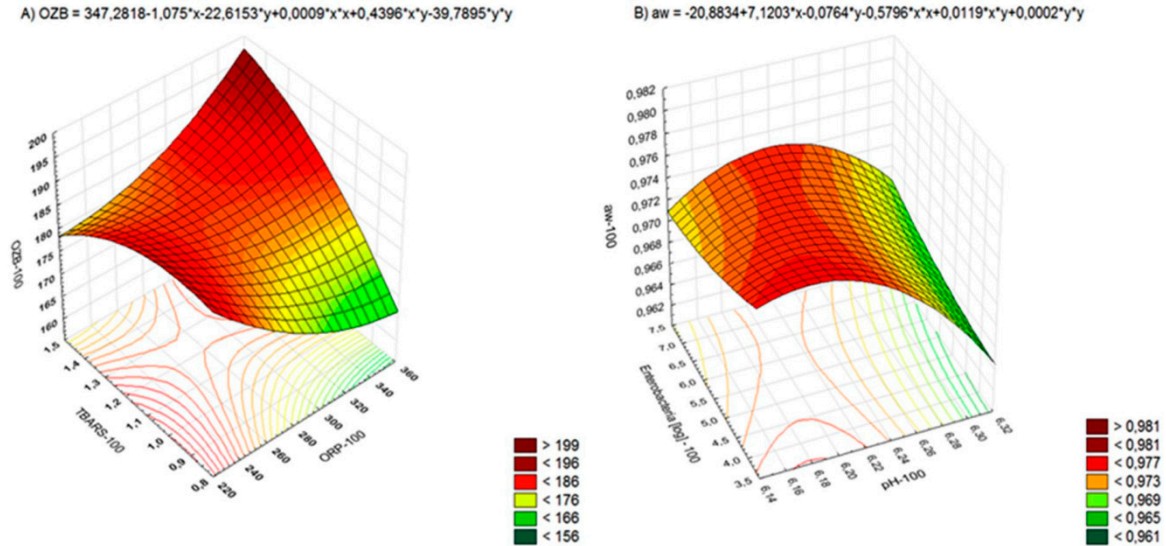

**Figure 6.** Mutual interactions of safety parameters for beef N_100: (**A**) Dependence of total heme pigment (OZB) on oxidation reduction potential (ORP; x) and thiobarbituric acid reactive substances (TBARS; y); and (**B**) Water activity ($a_w$) with respect to pH value (x) and number of *Enterobacteria* (y).

A decrease in the OZB content with an increase in ORP was observed. Higher ORP values indicate an intensification of oxidation processes in the sample. More oxidative metabolism causes color changes, leading to darkening of the meat. It is reflected in the decline in OZB values observed in this study. OZB was less susceptible to changes in the TBARS indicator (r = 0.213, $p > 0.05$). A relatively high content of OZB (<186) was observed almost in the entire TBARS range (0.8–1.5) specified in this study for the N_100 sample. Further, the increase in TBARS above 1.4 was associated with a decrease in OZB to <176.

## 4. Discussion

The increase in the number of micro-organisms while storage is one of the main agents that influences the quality of roasted, minced beef, leading to its spoilage as the consequence. It is crucial to provide meat products with a lower inclusion level of sodium nitrite with the same level of safety as that of products with the maximum level permitted.

Gonzalez and Diez [21] have shown the effectiveness of nitrite (50–150 ppm) in lowering *Enterobacteriaceae* count in Spanish sausage. A lower inclusion level of nitrite is needed to develop color than to control the amount of bacteria. Nitrite can also inhibit the growth of several aerobic and anaerobic microorganisms such as pathogen *C. botulinum* and contribute to the management of other microorganisms including *L. monocytogenes,* and *S. enterica serovar Typhimurium* [22]. Salt (NaCl), besides controlling *C. botulinum*, co-operates with nitrite and other factors such as acidity, meat type, heat treatment. It enables effectively control of the outgrowth of spores and thus widely helps in controlling *C. botulinum*. Salt at 5% (wt/vol) was indicated to completely inhibit *C. botulinum* growth in its optimal growth conditions [23]. Nitrite is able to inhibit bacteria more effectively at low pH or higher acidity [24]. The characteristics of curing with nitrite that make it an effective antibotulinal compound depend on the interactions of nitrite with several other variables. The variables that nitrite interacts with include salt, pH, heat treatment, spore level, nitrite input level during manufacture, and residual nitrite amount in the meat [25]. The characteristics of the competing flora, accessibility of iron in the product, and other supplements present in the meat including ascorbate, erythorbate, phosphate, etc. are other additional factors [26]. Furthermore, cooking, curing, and storage temperatures are the other important factors. In the literature, we found that the impact of nitrate on controlling *C. botulinum* toxin production was restricted—nitrite at the amount of 50 ppm resulted in two toxic samples out of 110 tested, and nitrite levels above the amount of 50 ppm resulted in zero toxic samples [27]. This particular observation is in accordance with Lövenklev et al. [28], who discovered that the amount of 45 ppm sodium nitrite effectively suppressed *C. botulinum* gene representation. Cui et al. [29] used a combination of sodium nitrite and spice extracts and found that sage extract inhibits the growth of *C. botulinum* and clove and nutmeg can inactivate bacteria when combined with 10 ppm of sodium nitrite. Christieansa et al. [30] showed that a 47% reduction in the nitrite concentration provided the same sanitary effect against *Salmonella* and *Listeria* as the regulatory inclusion level in French dry-fermented sausage. The obtained results indicated that the concentration of nitrite does not affect the LAB count. For LAB, the results agree with other studies showing that Micrococcaceae and LAB were not greatly affected by nitrate and nitrite [21,22].

A residual nitrite level from 10 to 15 ppm is recommended as a reservoir for the regeneration of cured meat color (30–50 mg/kg meat). Ahn and Maurer [31] recorded pinking effects in oven-roasted turkey breasts with the addition of as little as 1 ppm of sodium nitrite. Heaton et al. [32] also recorded almost the same results in cooked turkey rolls, chicken rolls, and pork shoulder rolls. Moreover, the authors reported that sensory panelists noticed pinkness or even pink color in turkey, chicken, and pork rolls for 2, 1, and 4 ppm sodium nitrite samples, respectively. The authors also pointed out that meat products with higher pigment concentrations (pork) needed higher nitrite levels for panelists to observe the visual pinking effects. Deda et al. [33] assessed the color parameters of frankfurters produced with different levels of sodium nitrite and tomato paste. The authors showed that the amount of sodium nitrite added to the frankfurters can be reduced from 150 to 100 ppm when

combined with 12% tomato paste without any negative effect on the quality parameter of the final product. Hayes et al. [34] proved that the amount of sodium nitrite could be reduced to 50 ppm when combined with 1.5% of tomato pomace powder with similar sensory qualities and microbiological stability compared to formulation with 100 ppm nitrite alone.

Another property of nitrite is its ability to retard the oxidation process during storage period and the subsequent warmed-over and rancid flavors developed during thermal processing of meat and meat products [35,36]. The antioxidant activity of nitrite is attributed to the potential of nitric oxide to bind to and stabilize heme iron of meat pigments during the meat curing process. Nitric oxide, being a free radical, can also trigger lipid autoxidation by chelate free radicals including peroxyl radicals. The nitrite binds free irons and stabilizes the heme iron, which can reduce lipid oxidation by limiting prooxidant activity of iron. The our data show the effect of different inclusion levels of sodium nitrite (from 50 to 150 mg/kg) and storage time on the oxidative stability of the roasted beef (Table 3, Figure 3).

Nitrite can form another antioxidant compound, for example, S-nitrosocysteine. Dethmers and Rock [37] stated that the addition of nitrite above the amount of 50 ppm to Thuringer sausage lowered the off-flavor development and bettered the flavor quality, whereas treatments with no nitrite added were believed to be the most rancid ones due to the poor flavor quality. Doolaege et al. [38] investigated the effects of different inclusion levels of sodium nitrite (40, 80, and 120 ppm) combined with different inclusion levels of rosemary extract (0, 250, 500, and 750 ppm) and reported that the concentration of sodium nitrite added to liver pate could be reduced to 80 ppm when rosemary extract is added at 250, 500, and 750 ppm concentrations without any negative effect on lipid oxidation and color parameters. A significantly higher ORP was observed for samples N_75 and N_50 than for samples N_150 and N_100 throughout the storage period. Samples treated with higher inclusion level of sodium nitrite (100–150 mg/kg) had decreased potential redox values compared to other samples by $\approx$18 mV (Table 3). Storage time (S) affected only ORP values ($p < 0.001$). In all study samples, a gradual decrease ($p < 0.05$) was shown at 7 days of storage, followed by a slight increase at 14 and 21 days of storage. According to Antonini and Brunoni [39], the ORP value tends to become lower at higher pH. In the our study, as pH decreased the ORP increased (Tables 1 and 3). The highest ORP values (N_75 and N_50) indicated that heme pigments in the ferric state had not stabilized the color of product [40], which confirmed the results of the ΔE showed on Figure 4.

The dependence of OZB on ORP was observed—higher ORP values is connected with intensification of beef oxidation. More oxidative metabolism causes color changes, leading to darkening of the meat [41]; this is reflected in the decline in OZB values observed in this study. OZB was less susceptible to changes in the TBARS indicator (r = 0.213; $p > 0.05$). A relatively high content of OZB (<186) was observed almost in the entire TBARS range (0.8–1.5) specified in this study for the N_100 sample. Further, the increase in TBARS above 1.4 was associated with a decrease in OZB to <176. Ineffective inhibition of the oxidation of meat ingredients is a serious threat to food safety in the opinion of many of the above-mentioned authors.

## 5. Conclusions

Nitrite is difficult to replace as a preservative, because it can simultaneously perform many functions. The present study showed the real possibility of reduction in the use of nitrite in meat products. The obtained results revealed that 100 mg/kg of sodium nitrite added would be sufficient for minced roasted beef, without significant effects on color, oxidative stability, and microbiological safety as compared to control (150 mg/kg). A reduction in the addition of sodium nitrate by half and two-thirds resulted in a decrease in pH and water activity. More importantly, the lower nitrite inclusion level (50 and 75 mg/kg) caused an acceleration of oxidation processes (higher TBARS values and ORP) in the product and color deterioration during 21 days of chilling storage.

Therefore, further studies are needed to investigate the complex effect of various reduced levels of nitrite and potential alternative compounds and/or technologies that can substitute nitrite.

**Author Contributions:** Conceptualization, K.M.W. and D.M.S.; Methodology, K.M.W., P.K., and D.M.S.; Validation, K.M.W., P.K., and D.M.S.; Formal Analysis, K.M.W. and D.M.S.; Investigation, K.M.W. and P.K.; Resources, D.M.S.; Writing—Original Draft Preparation, K.M.W. and D.M.S.; Writing—Review and Editing, D.M.S. and K.M.W.; Visualization, K.M.W. and D.M.S.; Supervision, K.M.W.; Project Administration, K.M.W.; Funding Acquisition, D.M.S.

**Funding:** This research received no external funding.

**Conflicts of Interest:** The authors declare no conflict of interest.

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
