# Peer review of "The Influence of Different Levels of Sodium Nitrite on the Safety, Oxidative Stability, and Color of Minced Roasted Beef"

_sustainability, doi:10.3390/su11143795_

Round 1

Reviewer 1 Report

The aim of this study was to collect actual data on the workable possibility of reducing the technological use of nitrites in the beef product according to the present trends in nutrition. The methodology was generally sound, and the study was well executed and well written. However, there are some general comments that might be addressed in the MS.

Tables and statistic outputs along the MS, are not very clear. Specifically, the way data (average) are presented in combination with the statistical analysis. The experimental design is based on a factorial model, working with interactions. Such a proper table should be organized in a column for the days (0, 7, 14 and 21) and in each column having the NaNO3 concentration, such letters in the values will easily show the differences among all columns. P values showing the effect of the factors and interaction will be at the end as is showed in this draft.

Figure 4. At time zero there is an increase in delta (150-100), which don't make sense. How is it possible to have this delta in color at time zero? Is this due to the variability and/or scarcity precision of the method to evaluate color?

Was this evaluated with T BAR ESSAY, please make it clear in the MS. what was the accuracy in the TBAR measuring?

Is it possible that this reduction in NaNO3 could interfere in the palatability of the final product -preserved meat?

Author Response

The response to the reviewer's comments is in attachment.

Reviewer 2 Report

The paper aimed to study the effect of different inclusion levels of sodium nitrite on safety, oxidative stability and color of minced roasted beef.

The specific comments are as follows:

Abstract

L19-20: Indicate the inclusion level of sodium nitrite.

L20: Consider to change “…was more vulnerable on lipid oxidation…” to “…was more vulnerable to lipid oxidation …”

L20: Consider rewriting the sentence “…The amount of minor products of lipid oxidation…”

L21: Consider to change “…to the sodium nitrite dose…” to “…to the sodium nitrite inclusion level …” , and throughout the manuscript.

L21-22: Consider to change “…The microbiological assessment did not present the existence of pathogenic bacteria after storage.…” to “…Clostridium spp., S. aureus and L. monocytogenes were not detected in any of the samples tested during all the experiment…”

 Introduction

L53: “…Today, nitrite is used to meet consumer requirements as to products that have extraordinary sensory characteristics and convenient merits connected with cured meats”.

Commentary: Nowadays, sodium nitrite is incorporated in this type of meat products because of safety reasons, mainly because of C. botulinum.

L74: Change “Nitrite inhibits the growth of food spoilage bacteria such as Salmonella enterica serovar Typhimurium, Listeria spp., and Clostridium botulinum …” to Nitrite inhibits the growth of pathogenic bacteria such as Salmonella enterica serovar Typhimurium, Listeria spp., and Clostridium botulinum…”

 Materials and Methods

L94: Please clarify:”… without quality defects…”

L96: Please clarify, how was the sodium nitrite incorporated (curing salt)? Indicate the product commercial reference.

L102: Please indicate how was the product temperature measured and the equipment used.

L104: Please indicate the characteristics of the packaging film and the packaging equipment used.

L109: Consider to change “The pH value and water activity were measured potentiometrically…” to “The pH value was measured potentiometrically…”

L113-115: Please clarify, the water activity was measured using the homogenate with water?

L132-133: Please indicate the characteristics of the equipment used.

L166-167: Please indicate how many samples were analyzed per each sodium nitrite inclusion level/storage time? The n value is missing in all tables.

Results

L178: Consider to change “Formulation (T) …” to “Sodium nitrite inclusion level (T)…”, and throughout the manuscript.

L189: Consider to change “Values are mean ± SD (range) …” to “Results are presented as mean ± SD …”, and throughout the manuscript.

L198: Consider to change “…A gradual increase observed in the water activity…” to “…A gradual increase was observed in the water activity…”

L228-229:” … the red nitrosyl-porphyrin ring system (often called nitrosomyochromogen) still exists even after heating to 120 °C.” Authors should introduce a reference to support this.

L242: Consider to change “The different levels of sodium nitrite have significant…” to “…The different levels of sodium nitrite had significant…”

L277: Correct the P value (P<0.00).

L280-281:”…Nitrite can form another antioxidant compound, for example, S-nitrosocysteine. In the present study, as pH decreased (Table 1), the ORP increased (Table 3).” is repeated.

L322: “…OZB was less susceptible to changes in the TBARS indicator (r = 0.213). Authors should indicate the significance level of the correlation found.

Discussion

L338-339: “…Salt at 5% (wt/vol) was indicated to completely inhibit C. botulinum growth in its optimal growth conditions.” In which meat product(s)? Please indicate a reference.

L384: Consider to change “…that the additive of nitrite above the…” to “…that the addition of nitrite above the …”

L394: Correct the P value (P<0.00).

L402-403: “…OZB was less susceptible to changes in the TBARS indicator (r = 0.213).” Authors should indicate the significance level of the correlation found.

 Conclusions

L417-418: “Therefore, further studies are needed to investigate the complex effect of various reduced levels of nitrite and potential alternative compounds and/or technologies that can substitute nitrite.”

Commentary: A consumer test would be very helpful.

Round 2

Reviewer 2 Report

The manuscript has been significantly improved.

Specific comment

L202: Correct the word “lvevel”

Author Response

The response to the reviewer's comments is in attachment.

Sustainability EISSN 2071-1050 Published by MDPI AG, Basel, Switzerland RSS E-Mail Table of Contents Alert
Back to Top